# Artificial Neural Network Modeling for Predicting and Evaluating the Mean Radiant Temperature around Buildings on Hot Summer Days

**Yuquan Xie** [1,2], **Wen Hu** [2,*], **Xilin Zhou** [1] , **Shuting Yan** [3] **and Chuancheng Li** [1]

1 School of Civil Engineering and Architecture, Wuhan University of Technology, 122 Luoshi Road, Wuhan 430070, China; xieyuquan7808@outlook.com (Y.X.); zhou.xilin@whut.edu.cn (X.Z.); licc@whut.edu.cn (C.L.)
2 Department of Architecture and Building Science, Graduate School of Engineering, Tohoku University, Sendai 980-8579, Japan
3 School of Civil Engineering and Architecture, Wuhan Institute of Technology, 693 Xiongchu Road, Wuhan 430205, China; yan.shuting@wit.edu.cn
* Correspondence: hu.wen.t4@dc.tohoku.ac.jp

**Abstract:** In recent years, the phenomenon of urban warming has become increasingly serious, and with the number of urban residents increasing, the risk of heatstroke in extreme weather has become higher than ever. In order to mitigate urban warming and adapt to it, many researchers have been paying increasing attention to outdoor thermal comfort. The mean radiant temperature (MRT) is one of the most important variables affecting human thermal comfort in outdoor urban spaces. The purpose of this paper is to predict the distribution of MRT around buildings based on a commonly used multilayer neural network (MLNN) that is optimized by genetic algorithms (GA) and backpropagation (BP) algorithms. Weather data from 2014 to 2018 together with the related indexes of the grid were selected as the input parameters for neural network training, and the distribution of the MRT around buildings in 2019 was predicted. This study obtained very high prediction accuracy, which can be combined with sensitivity analysis methods to analyze the important input parameters affecting the MRT on hot summer days (the days with the highest air temperature over 30 °C). This has significant implications for the optimization strategies for future building and urban designers to improve the thermal conditions around buildings.

**Keywords:** backpropagation algorithm; sensitivity analysis; outdoor thermal environment; mean radiant temperature; genetic algorithms

## 1. Introduction

The world is urbanizing faster and 68% of the population is expected to live in cities by 2050 [1]. The increased urbanization of the population, the urban heat island effect, and the effects of global warming have resulted in a rise in urban temperature [2–4]. According to research findings, it is estimated that the temperature increase will reach a peak of 1.5 °C between 2030 and 2050 [5]. Therefore, heatstroke is becoming a growing concern in cities. In Tokyo, ambulance evacuations due to heatstroke were six times greater than in previous years, and the number of patients suffering from heatstroke has been growing since 2010 [6], a trend that extends to other cities in Japan, including Sendai, roughly 300 km from Tokyo [7]. Therefore, urban warming is now becoming a serious problem, dramatically affecting outdoor thermal comfort and significantly increasing the risk of heatstroke for pedestrians [8]. To mitigate urban warming and adapt to it, effectively evaluating the urban thermal environment is the focus of current research, which will decrease human exposure to heat and the possibility of heat-related illness [9].

A growing body of literature [9–11] has evaluated the impact of urban warming using standard effective temperature (SET*), which is used to assess adaptation to urban

warming. As one of the key factors in calculating SET*, the MRT is the most important index that characterizes the effects of the thermal radiant environment on human thermal comfort [12,13]. When compared to standard meteorological indices such as air temperature, Thorsson et al. [14] propose that MRT might be a more appropriate indicator for evaluating intra-urban variances in thermal comfort conditions, particularly in a complex urban setting. So, the MRT has been extensively used in urban human-biometeorological investigations across the globe to parameterize levels of thermal comfort and heat stress during harsh weather [15–17].

Indeed, numerous research [18–22] indicates that climatic and building form elements such as solar location, date, sunlight duration, mean air temperature, and cloud cover may have an effect on the MRT. Shahrukh Anis et al. [23] found that sunshine duration could affect global solar radiation and further impact MRT and that the morphology of urban canyons will influence sunshine duration. Lindberg et al. [24] revealed that the shadows of buildings have the potential to limit incoming shortwave radiation and hence MRT, which is critical for MRT distribution. In recent years, several studies [25–27] have paid more attention to the sky view factor (SVF), which is an indirect representation of the built morphology; it has been demonstrated to have a significant impact on MRT and outdoor thermal environments. Dogan [28] indicated that MRT is impacted by the surface temperature of the adjacent urban features, particularly for pedestrians, and that the ground temperature has a stronger effect on MRT distribution than others [29]. Although the analysis of the parameters affecting MRT has focused mainly on weather or building parameters in recent years, if building-related parameters (SVF, building shadow, surface temperature, sunshine duration) are considered together with weather data in different urban morphologies, the order of importance of the various parameters to MRT in urban spaces can be quickly determined. Due to the difficulty of coping with the changing weather conditions, optimization solutions for building-related factors might be recommended for outdoor environment improvements.

Typically, sensitivity analysis approaches are employed to investigate the order of importance. However, there are two significant limits: (1) sensitivity analysis is used to evaluate the change of dependent variables when one of the independent variables is changed; in order to evaluate the order of importance of the parameters for MRT, a method for quickly and accurately calculating the MRT by the parameters is required. (2) There are numerous sensitivity analysis approaches, and it is vital to select one that is compatible with the quick calculation approach. Therefore, there is a need to develop a sensitivity analysis technique that is compatible with the quick calculation approach. Therefore, an appropriate prediction approach and sensitivity analysis approach for the prediction approach are critical for addressing these two concerns.

For prediction approaches, many studies have successively proposed different models that obtain weather data to make predictions [30]. Roman et al. [30] analyzed the application of different prediction methods, including artificial neural networks (ANN) [31–33], Gaussian process [34], polynomial regression [35], support vector machine [36], and linear regression [37] prediction methods, in building performance simulations by performing a literature review. According to the authors, the most extensively used approaches in building performance assessment are as follows: (accounting for 33%) is ANN, followed by polynomial regression methods (constituting 22.6%). Moreover, ANN methods are advantageous in some other fields due to their nonlinear mapping ability and favorable prediction data-processing ability, and many related building performance simulation studies have been performed over the past decade.

At present, some other ANNs have been utilized to overcome these problems. Kumar et al. [38] presented an energy analysis with the use of ANNs. Among those mentioned, a backpropagation neural network (BPNN), using a backpropagation algorithm, an enhanced version of the feedforward neural network (FFNN), was shown to be better at evaluating, estimating, and predicting a big dataset when compared to statistical approaches such as the least-squares method. Afterward, Wang et al. [39] determined that the

BP algorithm can be used with a linear ANN, which is a gradient descent algorithm that is widely utilized to train ANNs. Mohandes et al. [40] indicated that neural networks are the most extensively employed kind of network for prediction, in particular, the MLNN [41] and recurrent neural network (RNN). RNN has become one of the most popular neural networks in recent years, nonetheless, it is only useful for continuous-time prediction and data preparation [42].

However, BPNNs are limited in their ability to solve specific situations. Due to the fact that a BPNN corrects network connection weights using the root mean square error (RMSE) and gradient descent technique, unavoidable issues such as slipping into local minima, sluggish convergence speed, and overfitting may exist [43]. Kolhe et al. [44] used a hybrid prediction model that included a genetic algorithm (GA) and a backpropagation neural network (BPNN) to predict wind energy. Although the BPNN worked well, combining it with a GA resulted in more efficient and accurate outcomes. Then, Ata et al. and Meukam et al. [45,46] revealed that: (1) In regions including windy environments, the most often-utilized ANN is the MLNN; (2) Backpropagation algorithms seem to be the most frequently used algorithms for MLPNN training in wind energy conversion systems; and (3) GAs are among the most frequently used optimization algorithms for optimizing BPNNs. According to Zhu et al. [47], the backpropagation neural network model enhanced by the genetic algorithm (GA-BPNN) has greater accuracy than the BPNN method but a slower computation time. Consequently, the MLNN has higher accuracy than the one-hidden-layer FFNN. Therefore, the MLNN optimized by the GA and BP algorithm (MLNN-GABP) was selected to compare the prediction results of the MRT with the MLNN. The more accurate one could be used for sensitivity analysis.

In previous studies, numerous approaches have been presented to determine the relative relevance of ANN variables, such as the linkage weight method approach [48], the Garson algorithm approach [49], the partial derivative approach [50,51], and the sensitivity analysis approach [52]. Nonetheless, there is no unanimity due to the disparate priority assigned to the individual variables because the approach determines the relative relevance of the individual variable's method of sensitivity analysis [53]. As a result, a proper sensitivity analysis technique is required. Chan et al. [54] predicted outdoor thermal comfort using an MLNN with two hidden layers. For the summertime prediction, 12,500 training datasets were utilized and the R-value was 0.898, indicating its high accuracy. This study employs an MLNN that is capable of handling large amounts of data and achieving high accuracy and combines a biological and a weather-related metric to predict the predicted mean vote (PMV) and SET*, achieving both high prediction accuracy and ranking. According to this study, the sensitivity analysis approach was proven to be suitable for ANN. Therefore, the sensitivity analysis approach was chosen as the conference for analyzing the important variables for MRT in this study.

In this work, ANNs were used to predict the MRT distribution, and data were obtained every day in the summer from 2014 to 2019 in Sendai at noon true solar time (when the sun is at its greatest height and solar radiation is at its strongest). The main objective of this work is to develop MLNN-GABP models to predict the distribution of MRT at 1.5 m (pedestrian height) of the surrounding buildings. It was important to gather training and validation datasets for the ANN. The meteorological datasets from 2014 to 2018 were used as the inputs, whereas the MRT results calculated using the same weather dataset's settings were used as the target outputs. The validation approach used the 2019 meteorological datasets as the inputs for the trained neural network and the MRT results generated under the same weather dataset's settings as the validation outputs. The MRT prediction model was developed using a set of simplified-shape buildings in Sendai, Japan (38°16′06″ N, 140°52′10″ E).

The findings given in this paper are expected to aid in the development of a viable strategy for predicting MRT distributions without requiring extensive calculations. Urban planners and designers must provide pertinent recommendations and design measures based on optimal outdoor thermal environments.

## 2. Overview of This Study

In this study, the evaluation of the MRT is mostly accomplished by the collection of weather data, the development of neural networks, and sensitivity analysis. Weather data for the MRT calculation was collected from the Japan Meteorological Agency (JMA) [55]. A typical residential urban district consisting of simplified buildings in the latitude and longitude of Sendai was built in Japan, concerning previous research.

Firstly, the summer season period in Sendai must be defined, and weather data must be collected, primarily for the MRT calculations and as the input parameters to the neural network. The input for the MRT calculations and the training of the neural network will be the weather data recorded from the weather station. The expected output of this step will be the heat gain of each surface of the building during the summer season, and the MRT value around the building at a height of 1.5 m. With the trained neural network and the weather data for the whole summer period, a prediction for the MRT distribution will be made. The MRT distribution results and trained neural network will be sent for sensitivity analysis to calculate the order of importance of each input parameter to the MRT.

### 2.1. Weather Data Collection

Generally, the summer season is an extended period, and the weather conditions are different, including both rainy and sunny days. The different weather conditions also have a great impact on the results of the outdoor thermal environment. The three months of June, July, and August are considered summer in Japan, and the summer period differs by area. For each day, the JMA defines the days with a high temperature of over 25 °C as "summer days", and the days with a high temperature of over 30 °C as "hot summer days".

The JMA database was used for the MRT evaluation in this research. It is possible that the MRT readings in Sendai were impacted by factors such as global solar radiation and air temperature, as well as solar altitude, cloud cover, and the length of insolation. Predictive neural networks must be trained for at least four years in order to provide reliable findings, according to previous studies [56–59]. To train our neural networks, we used Sendai's meteorological data for the months of July and September from 2014 to 2018. From 2014 to 2019, Sendai experienced 126 hot summer days, according to the data. Additional computations and projections were made using hourly data.

### 2.2. Evaluation Model Determination

Furthermore, we took into account a plan floated up by Xuan [60] to use structures in the vicinity of an 865 m (x) by 785 m (y) plot as the basis for a created region with a 16 percent coverage rate (see Figure 1). On the basis of Xuan's prior work [60], a simpler model for residential areas was developed. The simulations in Sendai, depicted in Figure 1, were run using meteorological data and a model created with buildings facing each other (60 m (x) 15 m (y) 42 m (z)). At a z-position of 1.5 m, the simulation and assessment were carried out in the target area depicted in Figure 1b, which was 300 m (x) 300 m (y) in size. As shown in Figure 1c, the assessment area had a grid size of 3.75 m (x) 3.75 m (y). This approach has been utilized in prior research to evaluate outdoor environments [60].

An MRT calculation was done at noon true solar time in each example to calculate the 24 h heat balance. A modified urban block situated at Sendai's latitude and longitude was used for the simulations. Additionally, the MRT was computed for summer days from 2014 to 2019 in all grids. Based on the building's physical structure, the assessment area was split into 240 grids. Figure 1 shows the serial numbers. Building distances in the north-to-south direction were 30 m in the region investigated. The building's surface was separated into five sections: east, west, north, south, and the roof, in order to better understand how varied orientations, affect heat gain.

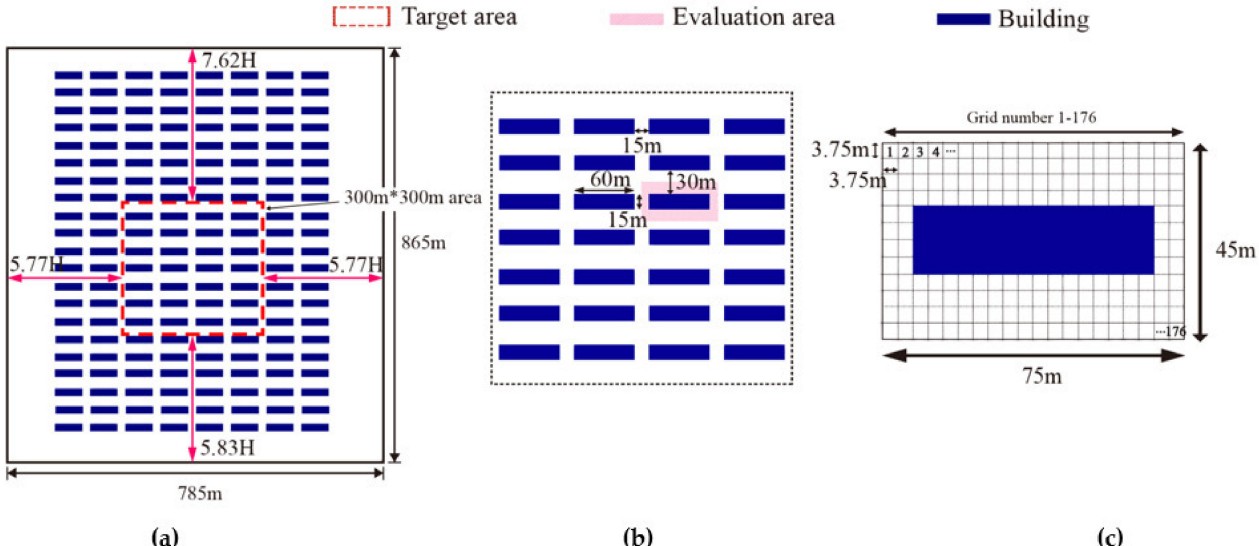

**Figure 1.** Simplified model of the residential area located in Sendai: (**a**) Residential area; (**b**) target area; (**c**) evaluation area and grid number.

## 3. Methodology

### 3.1. Heat Transfer Analysis and MRT Calculation

All surfaces in the computational domain were separated into tiny surfaces for the purpose of calculating urban surface temperatures. Direct solar radiation, sky radiation, and longwave radiation between the surface and adjacent surfaces were all analyzed for each surface. The form factor (i.e., configuration factor or view factor) was computed using the Monte Carlo technique [61,62], and the radiative heat transfer was determined using the Gebhart absorption coefficient [63]. Given that the Gebhart absorption factor varied with absorptivity as a function of shortwave and longwave radiation, we estimated these changes independently for each surface. As previously shown [64–66], metropolitan surface temperatures are well-recreated when the aforementioned heat exchanges are included. Figure 2 shows the structures of the ground and buildings (wall and roof), and Table 1 lists the corresponding thermal properties, which are both the same as the environment of the previous study [60]. The resources were obtained by using the parameters of real building materials in the guidelines from Sendai, Japan [67].

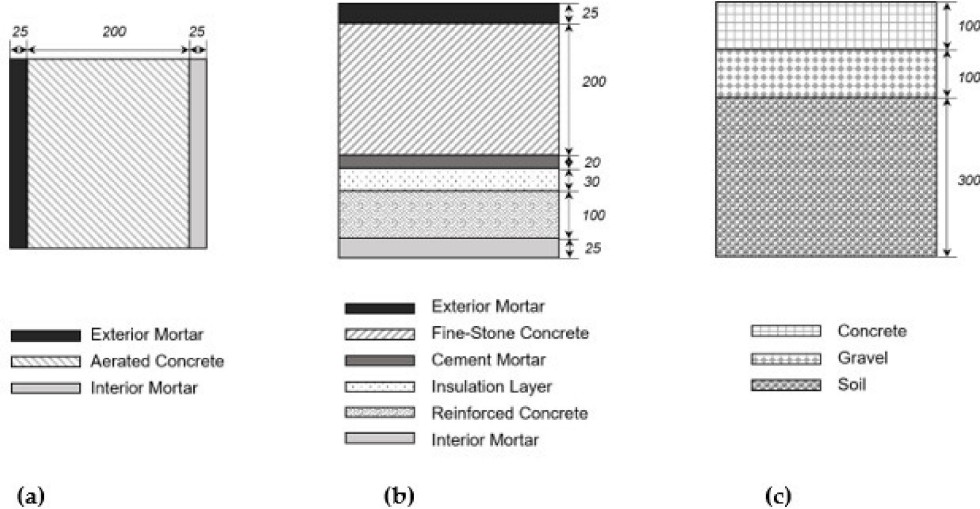

**Figure 2.** Structures of the ground and buildings (wall and roof): (**a**) Building wall; (**b**) Roof; (**c**) Ground.

**Table 1.** Structures and thermal properties of the ground, wall, and roof.

|  | No. of Layers | Material for Each Layer | Thickness (mm) | Thermal Conductivity (W/(m·K)) | Specific Heat per Unit Volume (kJ/m³·K) |
|---|---|---|---|---|---|
| entry 1 | 1 | Concrete | 100 | 1.28 | 1900 |
|  | 2 | Gravel | 100 | 0.62 | 1500 |
|  | 3 | Soil | 300 | 1.50 | 3100 |
| entry 2 | 1 | Exterior mortar | 25 | 0.93 | 1890 |
|  | 2 | Aerated concrete | 200 | 0.22 | 735 |
|  | 3 | Interior mortar | 25 | 0.87 | 1785 |
| entry 3 | 1 | Exterior mortar | 25 | 0.93 | 1890 |
|  | 2 | Fine stone concrete | 40 | 1.51 | 2116 |
|  | 3 | Cement mortar | 20 | 0.93 | 1890 |
|  | 4 | Insulation layer | 30 | 0.036 | 41.4 |
|  | 5 | Reinforced concrete | 100 | 1.74 | 2300 |
|  | 6 | Interior mortar | 25 | 0.87 | 1785 |

Through the use of radiation modeling, we were able to determine the net radiation heat gain of the building's outside surfaces. The three-dimensional multi-reflections of shortwave and longwave radiations were taken into account in order to achieve excellent prediction accuracy in this study. The Monte Carlo approach was used to compute the shape factor (also known as the configuration factor or view factor), and the Gebhart absorption factor was used to quantify the radiative heat transfer between the two surfaces. In this calculation, the radiation simulation was run every 10 min, and the output data were recorded every one hour. In the method by Nakamura [68], the human body is approximated as a rectangular prism (shown in Figure 3). Based on Nakamura's research, previous researches [9,10,60,69] have shown that when the heat balance calculation is used, urban surface temperatures are well-reproduced. Associated with the formulas in these literatures, the same code is adopted and run in the Fortran environment to calculate the MRT in this study. Equation (1) shows the MRT calculation method from Ref. [60].

$$\sigma T_{mrt}^4 = \sum_{l=-3}^{3} q_l \alpha_h c_l + \sum_{l=-3}^{3} \left( \sum_{j=1}^{n} B_{lj} \sigma T_j^4 c_l \right) \tag{1}$$

where $T_{mrt}$ is the MRT (K); $c_l$ is the weighting factor for surface l of a rectangular prismatic human body, which is 0.024 for the top and bottom surfaces, and 0.238 for the lateral surfaces; $l$ is the surface index of a rectangular prismatic human body, $l$ is 1, 2, 3 or −1, −2, −3; $q_l$ is the amount of shortwave radiation (direct, sky, and diffuse solar radiation) at surface $l$ of a rectangular prismatic human body (W/m²); $\alpha_h$ is the shortwave radiation absorption coefficient for the human body, $\alpha_h = 0.5$; $T_j$ is the temperature of the surfaces surrounding the rectangular prismatic human body (K); $B_{lj}$ is the Gebhart absorption coefficient, which is from the rectangular prismatic human body's surface l to surface $j$; and $\sigma$ is $5.67 \times 10^{-8}$ W/(m²·K⁴), the Stefan–Boltzmann constant [58].

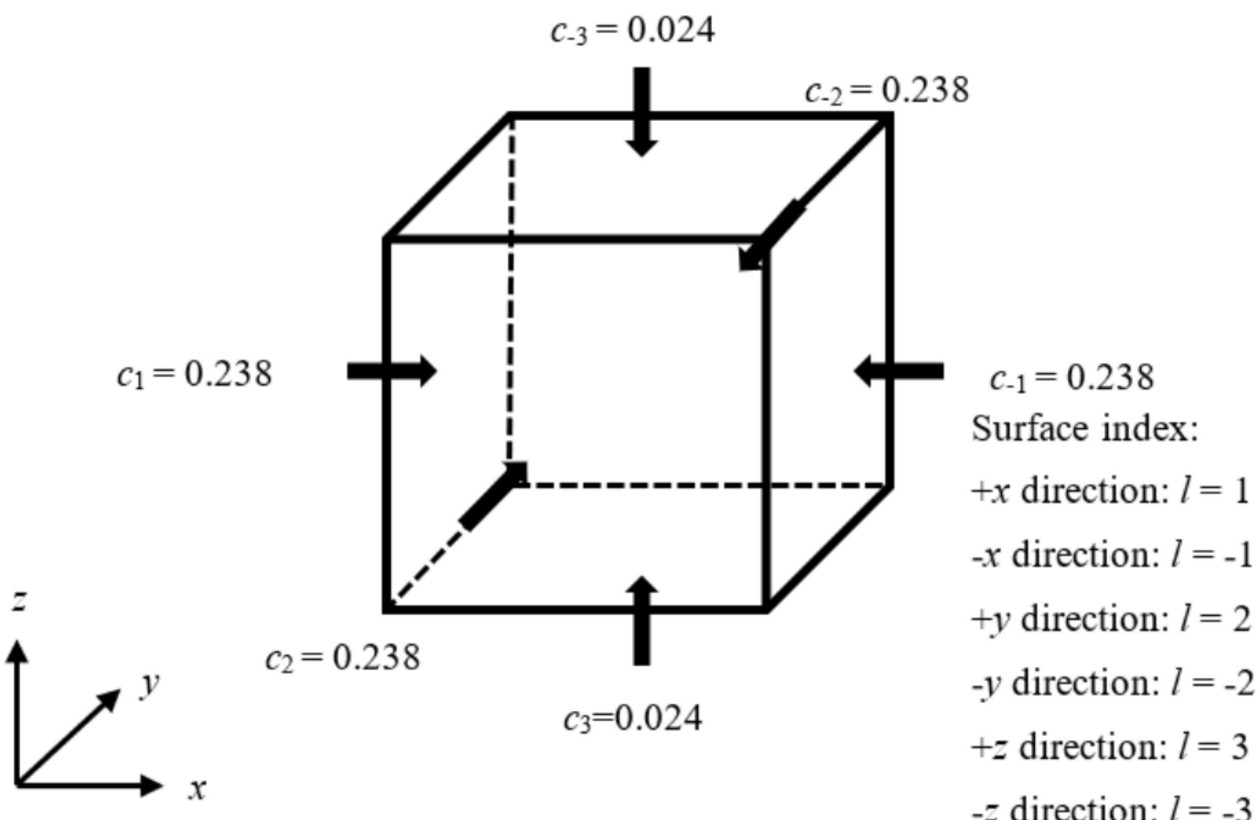

**Figure 3.** Rectangular prismatic human body.

### 3.2. Multi-Layer Neural Network Optimized by the Backpropagation Algorithm and GA Method

In recent decades, multi-layer neural networks have become widely used. Multi-layer neural networks [70] have received a lot of attention and have been effectively used in a variety of fields, including combinatorial optimization. FFNN neural networks are used to optimize both networks, which comprise an input layer, a hidden layer, and an output layer, with the input layer's neurons feeding parameters into the hidden nodes. When it comes to multivariate input, MLNN has an advantage over FFNN since it can be built with several hidden layers. Backpropagation is an algorithm that requires many occurrences of forward propagation in order to optimize its prediction outcomes, which was first suggested by Rumelhart [71]. A key feature of the backpropagation algorithm is the fact that the signal is transferred from the input to the output, whereas the error is conveyed from the output to the input. There are three layers in the backpropagation-optimized neural network: the input, the implicit, and the output. Using the direction propagation algorithm, the neural network may more easily achieve its learning objective by improving the connection methodology, connection weights, activation mechanism, and threshold of each neuron in each layer. The structural model is shown in Figure 4.

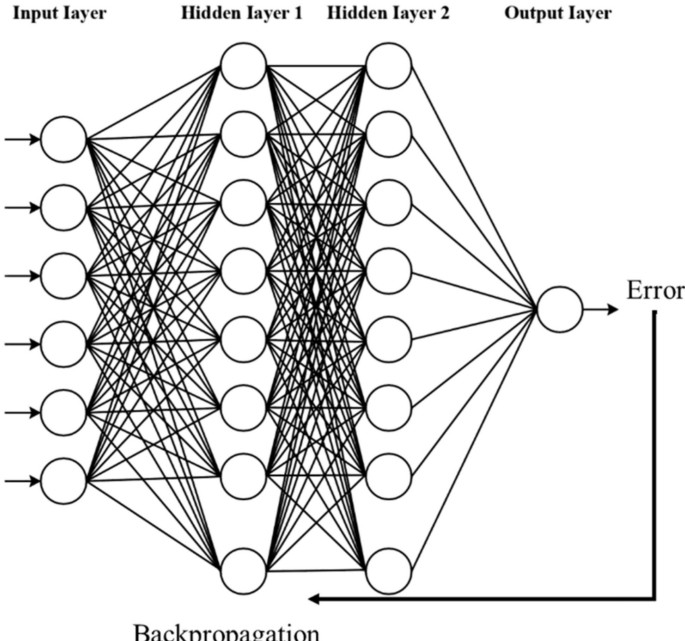

**Figure 4.** The structure of multilayer neural network optimized by backpropagation algorithm.

To keep the error within the specified range, the weights and thresholds can be corrected using the gradient descending approach. Equations (3)–(6) demonstrate the techniques for adjusting the weights of the hidden layer–output layer and the input layer–hidden layer; this is a kind of multilayer feedforward network trained using an error backpropagation algorithm that propagates errors to the weights and biases [71,72]. The error between the expected and actual outputs Ep (variance is used as an example in this study) is indicated in Equation (2).

$$Ep = \frac{1}{2} \sum_{j=0}^{m-1} (y_j - d_j) \tag{2}$$

Amount of change in weights $\Delta W$ can be calculated by:

$$\Delta W = -\eta \frac{\partial Ep}{\partial w} \tag{3}$$

Amount of change in thresholds $\Delta b$ can be calculated by:

$$\Delta b = -\eta \frac{\partial Ep}{\partial b} \tag{4}$$

Weights $b_{old}$ will be updated to $b_{new}$ by:

$$b_{new} = b_{old} - \Delta b \tag{5}$$

Weights $W_{old}$ will be updated to $W_{new}$ by:

$$W_{new} = W_{old} - \Delta W \tag{6}$$

where *E* represents the mean square error and outj denotes the *j*th neuron output. $\eta$ is the learning rate parameter that determines the network's stability and rate of network convergence; $\eta$ is a constant ranging between 0 and 1 that has been set to 0.01 in this study. *y*, *d*, *w*, *b*, and *Ep* are predicted output, actual output, weights, biases, and variance, respectively.

The neural network optimized using the BP method is likewise limited to some extent. In the first place, the BP neural network's application impact is greatly affected by the initial weighted value and the threshold value. This leads to the neural network being stuck in local minima during forward propagation, reducing the prediction performance; secondly, BP neural networks employ gradient descent, requiring an excessive amount of training time to optimize a complex objective function. As a result, the algorithm takes longer to reach its final output and requires more iterations. Genetic algorithm (GA) is used in this research to improve the weighted value and threshold value of the BP neural network and to create a new GA-BP neural network algorithm [45,73,74].

Steps for establishing and training a neural network model using Matlab's GA and BP neural network prediction model are as follows:

(1) Remove all the environment variables and normalize the data. The population sample's weighted value and threshold value are computed using the BP neural network. y and o represent the desired and expected outputs, respectively [75],

$$F = k\left(\sum_{i=1}^{m} \text{abs}(y_i - o_i)\right) \tag{7}$$

where $m$, $y_i$, and $o_i$ represent the number of neurons in the output layer, the $i$th expected value, and the $i$th predicted value, respectively.

(2) Create a new feedforward neural network, with a learning rate of 0.05, and set training parameters such as the factor of momentum, minimum root mean square error for the training, and the prediction of the BP network.

(3) Initialize genetic algorithm parameters. Confirm the number of iterations to be 50, the population size to be 10, the crossover probability to be Pc = 0.75, and mutation probability to be Pm = 0.25.

(4) Initialize the population. A random population of chromosomes is used to record the best and average fitness of the chromosomes in each generation of the iteration. Individuals with outstanding fitness are chosen to carry on to the next generation. In this research, the roulette approach is used to determine fi, which reflects the fitness of the individual i and can be computed by Equation (8),

$$f_i = k / F_i \tag{8}$$

$p_i$ denotes the likelihood of an individual $i$ being selected; it may be determined using Equation (9),

$$p_i = \frac{f_i}{\sum_{i=1}^{N} f_i} \tag{9}$$

where $F_i$ is the fitness function of the individual $i$; $k$ is the selectivity coefficient.

(5) Solve the best initial threshold value and weighted value through iteration. Through a process of selecting, crossing, and variating the population, it is possible to determine the population's minimum and maximum fitness chromosomes and their relative positions, and then use those values to replace the previous generations' worst individuals and their corresponding fitness values. This process continues until the maximum number of iterations has been reached. During the calculating procedure, the GA algorithm uses the entity code. In this study, the real number crossing method is used, where $a_{xi}$ represents the bit $i$ of the $x$th chromosome, which can be calculated by Equation (10)

$$a_{xi} = a_{xi}(1 - b) + a_{yi}b \tag{10}$$

$a_{yi}$ represents the bit $i$ of the $y$th chromosome, it can be calculated by Equation (11),

$$a_{yi} = a_{yi}(1 - b) + a_{xi}b \tag{11}$$

where $b$ represents the random number ($0 \leq b \leq 1$), respectively. Based on the formula below, there is a definite possibility that when one individual is selected, certain genes are transformed into other alleles:

$$a_{ij} = \begin{cases} a_{ij} + (a_{ij} - a_{\max}) \cdot f(g), r > 0.5 \\ a_{ij} + (a_{\min} - a_{ij}) \cdot f(g), r \leq 0.5 \end{cases} \tag{12}$$

Among Equation (12), the iteration formula is shown below:

$$f(g) = r_2 \left(1 - \frac{g}{G_{\max}}\right)^2 \tag{13}$$

where $a_{\max}$, $a_{\min}$, $r_2$, $g$, $G_{\max}$, and $r$ represent the upper bound of the gene $a_{ij}$, the lower bound of the gene $a_{ij}$, random number, iteration times, maximum times, and random number ($0 \leq r \leq 1$), respectively.

(6)    In order to train and predict the GA-BP network, the BP network should be assigned the ideal initial weighted value and the threshold value.

### 3.3. Sensitivity Analysis Approach

When Chan et al. [54] combined the predictions of neural networks in order to derive the order of importance for various types of input parameters, they used a sensitivity analysis method to demonstrate how to account for both the influence of weather parameters and also other related parameters on the outdoor thermal environment. Thus, the same sensitivity analysis technique is used in this research to determine the relative relevance of various kinds of input parameters, yielding the same conclusion as in the prior study. The approach employed in this investigation is as follows [76]. The following formula was used to obtain the percentage band for the input variables:

$$\frac{Xi_{\max} - Xi_{\min}}{100} \tag{14}$$

$X_i$ stands for the $i$th input variable. In the ANN model, a 4% band (+2% and −2%) of each input variable was trained. Before training, the values of all the input variables become $Y_i + 2\% X_i$ and $Y_i - 2\% X_i$. After that, the differences ($\Delta yi$) between $Y_i + 2\% X_i$ and $Y_i - 2\% X_i$ were calculated. For each variable, the index of the order of importance ($I_{Ri}$) for each variable was determined by:

$$\frac{\Delta yi}{Yi_{\max} - Yi_{\min}} \tag{15}$$

After that, the input variables were classified according to their significance. As shown in the References [76–79] section, a succession of ANNs was employed to highlight the significance of certain components in a particular order. The employment of a collection of ANNs has been found to be promising in terms of producing relatively consistent results while avoiding the instability problem associated with a single optimum network [80]. As a consequence, rather than using a single ideal network, this research made use of collections of neural networks. The relevance of the input variables was assessed using sets of trained neural networks. For additional examination, a total of twenty of the strongest neural models from winter and summer were selected. The R and MSE values from the validation datasets were utilized to assess the formed ANN model's prediction performance. Following that, a sensitivity analysis was conducted of all neural networks, and a ranking of each neural network's order of significance was obtained.

## 4. The Structure of the Neural Network

### 4.1. Selecting the Input Parameters

Correlation coefficients were calculated between each pair of input variables using weather data from JMA that could affect MRT, including solar direction, solar altitude, air temperature at noon true solar time, daily average air temperature, insolation time, minimum air temperature, global solar radiation, maximum air temperature, and cloud cover. The related parameters in each grid that might affect the MRT and that were proven in previous studies were selected for the correlation coefficient analysis. Among them, the sky view factor of each grid, sunshine duration before noon true solar time, and the shadow ration in each grid at noon true solar time are related to the morphology of urban space. These three parameters can be calculated in SketchUp and its plug-ins. The ground surface temperature has been shown to impact the MRT at the pedestrian level, and this may be determined using the previously described heat balance analysis. The majority of the correlation coefficients between the input variables are shown in Table 2. There are significant collinearities detected, with correlation values greater than 0.8 [54]. A strong correlation between the input variables may result in the ANN models being overfit and introducing many redundancies.

**Table 2.** Correlation coefficients between the input variables.

|     | X1 | X2 | X3 | X4 | X5 | X6 | X7 | X8 | X9 | X10 | X11 | X12 | X13 |
| --- | --- | --- | --- | --- | --- | --- | --- | --- | --- | --- | --- | --- | --- |
| X1 | 1 | | | | | | | | | | | | |
| X2 | 0.740 | 1 | | | | | | | | | | | |
| X3 | 0.280 | 0.036 | 1 | | | | | | | | | | |
| X4 | 0.342 | 0.051 | 0.604 | 1 | | | | | | | | | |
| X5 | 0.471 | 0.078 | 0.421 | 0.931 | 1 | | | | | | | | |
| X6 | 0.356 | 0.068 | 0.608 | 0.907 | 0.940 | 1 | | | | | | | |
| X7 | 0.557 | 0.099 | 0.104 | 0.741 | 0.907 | 0.743 | 1 | | | | | | |
| X8 | 0.192 | 0.153 | 0.946 | 0.689 | 0.515 | 0.691 | 0.221 | 1 | | | | | |
| X9 | 0.096 | 0.073 | 0.796 | 0.403 | 0.245 | 0.415 | 0.036 | 0.884 | 1 | | | | |
| X10 | 0.182 | 0.079 | 0.116 | 0.049 | 0.118 | 0.105 | 0.084 | 0.031 | 0.059 | 1 | | | |
| X11 | 0.221 | 0.147 | 0.173 | 0.289 | 0.214 | 0.326 | 0.450 | 0.038 | 0.176 | 0.032 | 1 | | |
| X12 | 0.178 | 0.143 | 0.116 | 0.004 | 0.071 | 0.059 | 0.151 | 0.157 | 0.018 | 0.031 | 0.046 | 1 | |
| X13 | 0.096 | 0.116 | 0.004 | 0.071 | 0.059 | 0.151 | 0.157 | 0.018 | 0.132 | 0.031 | 0.046 | 0.074 | 1 |

X1: solar altitude, X2: solar direction, X3: insolation time, X4: air temperature at noon true solar time, X5: daily average air temperature, X6: maximum air temperature, X7: minimum air temperature, X8: global solar radiation, X9: cloud cover, X10: the sky view factor of each grid, X11: sunshine duration before noon true solar time, X12: the shadow ration in each grid at noon true solar time, X13: ground surface temperature.

Several factors that may have had an impact on the MRT between 2014 and 2019 were studied for correlation and are shown in Table 2. The average air temperature and the average air temperature at noon true solar time, as well as the maximum and minimum air temperatures, all have strong relationships, as shown in Table 2. Because of this, there were two viable options: one required picking just the average air temperature, and the other required selecting both the air temperature at noon true solar time as well as the lowest air temperature. The second choice was selected due to the need for as many diverse input variables as was reasonably possible. Moreover, as the length of insolation and the amount of cloud cover were substantially connected with solar radiation and with one another, global solar radiation was selected as the input parameter for the training process. As a consequence, there is little association between the grid metrics associated with each grid point and the meteorological data.

Accordingly, average air temperature, maximum air temperature, insolation duration, and cloud cover were removed and replaced with minimum air temperature, air temperature at noon true solar time, and global solar radiation. Therefore, solar altitude, solar direction, air temperature at noon, minimum air temperature, global solar radiation, the sky view factor of each grid, sunshine duration before noon true solar time, the shadow ration in each grid at noon true solar time, and ground surface temperature can be used for training neural networks as the input parameters.

### 4.2. The Structure of MLNN-GABP

According to the findings of this study, the database was initially divided into two groups: one containing the weather and building-related parameters from 2014 to 2018 as well as the MRT of each grid from their simulation results as the training dataset (which contained 80 percent of the data), and another containing the weather and building-related parameters from 2019 along with the MRT of each grid from their simulation results as the validation dataset (which contained 20 percent of the data). In the training set, three subsets are used to determine performance: 60 percent is used for training, 20 percent is used for validation, and 20 percent is used for testing. Both the validation and test datasets are constructed in such a way that overfitting on the training data is avoided, resulting in a more realistic accuracy of the test data. It seems that the neural network was effectively trained when the training set, validation set, and test set all provide appropriate training results, suggesting that it was successfully trained.

As a result of a broad variety of training environments and a wide range of data, a technique for normalizing the data is needed. Oversaturation, which occurs when bigger datasets cover smaller datasets, slows down mechanical learning, hence standardized data is often utilized to train data in a consistent range. For predicting outdoor thermal comfort, the following standardization approach was used in the literature, and the current research makes reference to that method by applying the formula:

$$NV = \frac{NV_{\max} - NV_{\min}}{V_{\max} - V_{\min}} \times (V - V_{\min}) + NV_{\min} \tag{16}$$

where $NV$ is the input or output normalized vector; $NV_{\max} = 1$; $NV_{\min} = -1$; $V$ is the input or output data; $V_{\max}$ is the maximum of the input or output data; and $V_{\min}$ is the minimum of the input or output data.

The structure of MLNN-GABP is shown in Table 3. The validation dataset comprises 25 hot summer days in 2019 (a total of 25 days times 176 grids = 4400 validation data) and is used to validate the validation set. In the training step, all 176 hot summer days from 2014 to 2018 were selected as the training dataset (a total of 114 days times 176 grids = 20,064 validation data), and the ratio of the training set, test set, and validation set is 3:1:1 in the training step. In the test step, the ratio of the training set, test set, and validation set is 3:1:1. The input parameters in each grid are comprise mostly weather parameters, with a total of nine input parameters in total for each grid. The number of hidden layers was determined by the application of Equation (17). At a height of 1.5 m, the output parameter is the mean solar time (MRT) at midday true solar time for each grid. Table 3 lists the variables that are used to construct the input layer, hidden layer, and output layer, respectively. The algorithms for the neural networks, genetic algorithms, and backpropagation algorithms are all implemented in MATLAB using the code that was developed for each method. The multilayer feedforward neural networks that we used for this investigation were created using MATLAB-based software, and the ANN models were tested using the results.

$$2 \times \sqrt{N_{input} + N_{output}} \leq N_{hidden} \leq 2 \times N_{input} + 1 \tag{17}$$

where $N_{input}$ is the number of input neurons, $N_{output}$ is the number of output neurons, and $N_{hidden}$ is the number of hidden neurons.

**Table 3.** ANN model structure.

| Prediction Period | Random Validation |
|---|---|
| Input layer | Number of neurons: 9 |
| Weather parameters | Solar altitude<br>Minimum air temperature<br>Global solar radiation<br>Solar direction<br>Air temperature at noon true solar time |
| The related parameters in each mesh | Sky view factor of each grid<br>Sunshine duration before noon of the true solar time<br>Shadow ration in each grid at noon true solar time<br>Ground surface temperature |
| Hidden layer | Number of neurons: 7–19 |
| Number of training cases | 20,064 |
| Number of validation cases | 4400 |
| Output layer | Number of neurons: 1 (MRT) |

To validate the suggested technique, we compared the MLNN and MLNN-GABP outcomes. The root mean square error (*RMSE*) and mean absolute percentage error (*MAPE*) were employed to assess these three models' predictive accuracy. The root mean square error (*RMSE*) values were determined using:

$$MSE = \sqrt{\frac{1}{n}\sum_{i=1}^{n}(x_i(t) - y_i(t))^2} \tag{18}$$

and the *MAPE* values were calculated using

$$MAPE = \frac{1}{n}\sum_{i=1}^{n}\left|\frac{x_i(t) - y_i(t)}{x_i(t)}\right| \times 100\% \tag{19}$$

where $x_i(t)$ and $y_i(t)$ represent the *i*th actual and predicted values at time *t*, respectively.

MAPE was employed to assess predictive performance in the previous research: MAPE is smaller than 10% (excellent), MAPE is between 10% and 20% (acceptable), MAPE is between 20% and 50% (fair), and MAPE is smaller than 50% (bad).

## 5. Validation Results

### 5.1. Validating the Input Parameters and Determining Hidden Layers

By putting the training through its paces, and the validation and test sets of ANNS, the prediction performance was tested. As shown in Table 4 below, increasing a parameter implies that the trained neural network is not overfit and that the training parameters are reasonable. After incorporating the weather, shadow, sunshine, and sky view factors, the prediction accuracy was significantly improved, and after including the ground surface temperature in each grid, the prediction results were even more accurate, with an R-value greater than 0.999, indicating extremely high accuracy.

The R-values of the training, validation, and test sets with varied hidden layer counts, as well as the predicted RMSE and MAPE, were evaluated in this study. For the purposes of validating and comparing the prediction performance with the various hidden layer count, Table 5 shows the R-values, RMSE, and MAPE in the six MLNN-GABP structures with the smallest RMSE. Structured with 9 input layers, 15 hidden layers, 12 hidden layers, and 1 output layer, it indicates very high prediction accuracy, with an RMSE of 0.13, and a MAPE of 0.23%.

**Table 4.** A summary of R-values of the subset of the training dataset and validation dataset after adding additional variables.

| Added Variable | R-Values of the Training Set | R-Values of the Validation Set | R-Values of the Test Set |
|---|---|---|---|
| Weather parameters | 0.6732 | 0.6542 | 0.6643 |
| + Sky view factor of each grid, | 0.7556 | 0.7589 | 0.8013 |
| + Sunshine duration before noon true solar time, | 0.8059 | 0.8011 | 0.8213 |
| + Shadow ration in each grid at noon true solar time | 0.93478 | 0.9365 | 0.9372 |
| + Ground surface temperature | 0.99986 | 0.9998 | 0.9997 |

**Table 5.** The parameters of ANNs used for determining the hidden layer.

| No | ANN Structure | R-Value | | | RMSE | MAPE |
|---|---|---|---|---|---|---|
| | | Training Set | Validation Set | Test Set | | |
| 1 | 9-15-12-1 | 0.99986 | 0.9998 | 0.99975 | 0.13 | 0.23% |
| 2 | 9-13-11-1 | 0.99752 | 0.99864 | 0.99741 | 0.15 | 0.28% |
| 3 | 9-16-12-1 | 0.99652 | 0.99698 | 0.99701 | 0.17 | 0.34% |
| 4 | 9-17-12-1 | 0.99521 | 0.99523 | 0.99710 | 0.19 | 0.45% |
| 5 | 9-15-10-1 | 0.99465 | 0.99501 | 0.99601 | 0.21 | 0.48% |
| 6 | 9-16-11-1 | 0.99425 | 0.99465 | 0.99532 | 0.24 | 0.57% |

## 5.2. Comparison of MLNN-GABP and MLNN

To evaluate the prediction performance of the GA and BPs' effect on neural network training, Figure 5 compares the prediction results of MLNN with the prediction results of MLNN-GABP, and also shows the prediction performance of the training dataset, validation set, and test set. The R-value of the training dataset does not differ significantly from the R-values of the validation and test sets, indicating that the training performed is not overfitting. According to the training results in this study, both neural networks have a good training structure (R-value greater than 0.90), but the results of the MLNN-GABP optimization are obviously more accurate than MLNN. When the neural network without the GA and BP algorithm has a greater error at some points and the accuracy of the training results is improved by the GA and BP algorithm, there are smaller errors in individual samples.

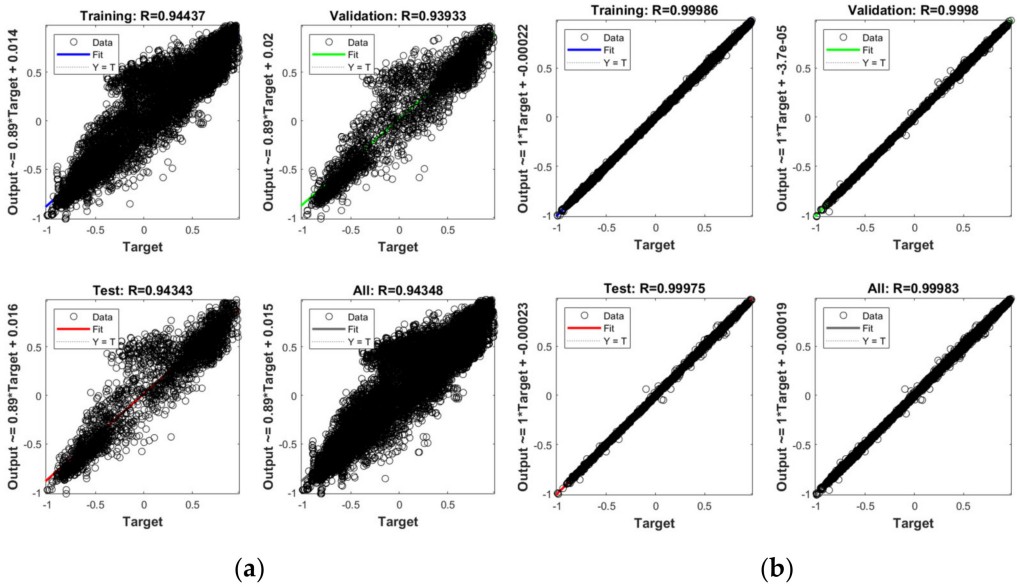

**Figure 5.** The training results of different neural networks: (**a**) MLNN; (**b**) MLNN-GABP.

### 5.3. The Prediction Results of MRT Distribution

These findings are depicted in Figure 6, which shows that the MRT values predicted by MLNN-GABP are exceptionally accurate in hot summers. The RMSE and MAPE values of 0.13 °C and 0.23%, respectively, show that the predicted and calculated values almost coincide, suggesting an extraordinarily high degree of accuracy. Using RMSE and MAPE, the predicted results are compared by days in Figure 7. As can be observed, the RMSE for all predicted days is less than 0.2 and the MAPE is less than 0.3%, indicating daily extremely high prediction accuracies, with 13 August having the highest error and 7 August having the lowest error. MRT distribution plots were generated for the three days with the highest and lowest errors, and those closest to the average error, as shown in Figure 8. When the predicted and calculated values were compared, just a 1 °C discrepancy was found, and the distribution plot results were identical.

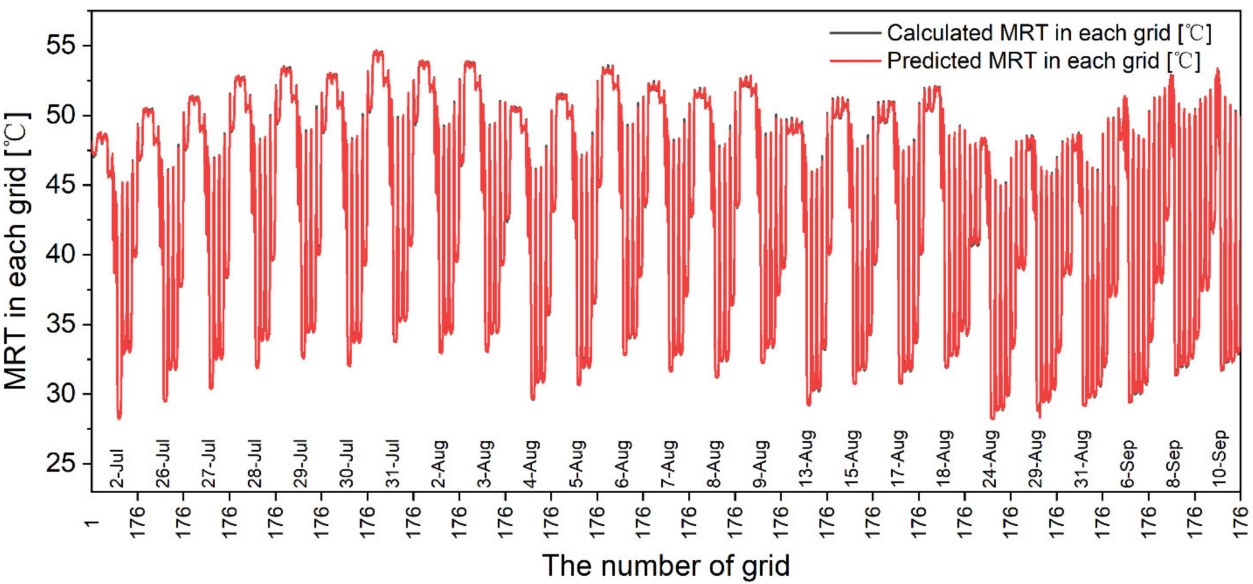

**Figure 6.** The validation results on the MRT distribution in each mesh on hot summer days in 2019.

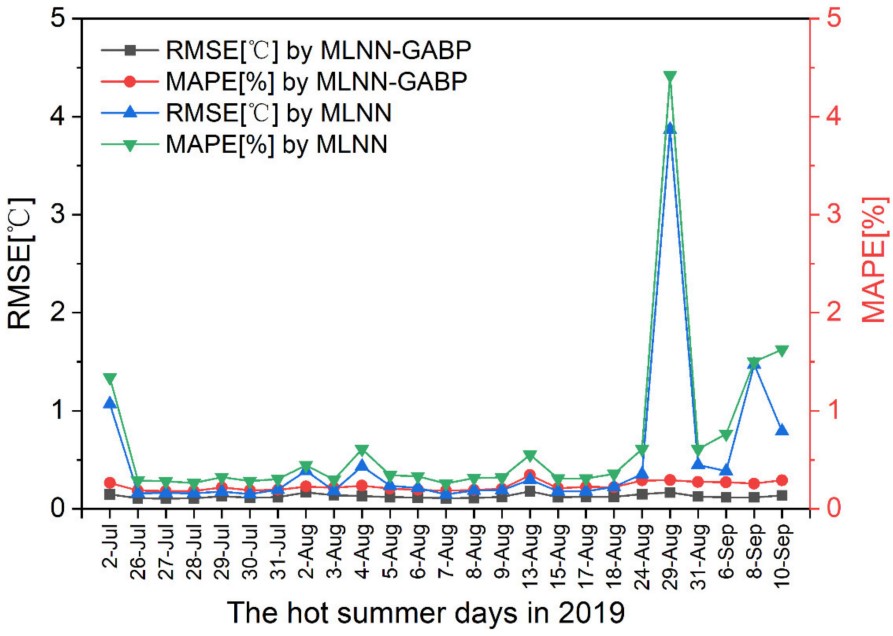

**Figure 7.** The RMSE and MAPE of the prediction results in each mesh on hot summer days in 2019.

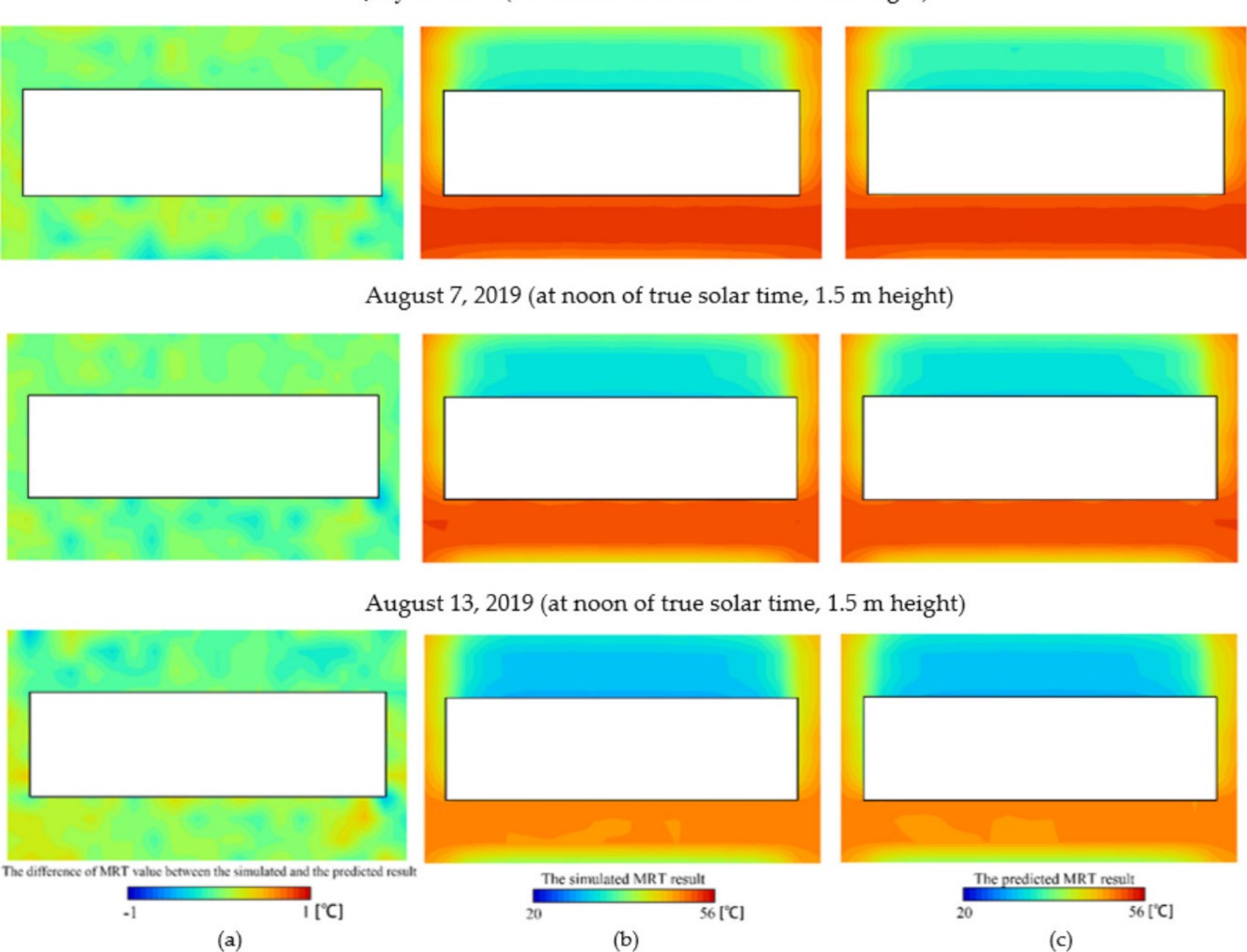

**Figure 8.** Comparison of the predicted results and calculated results: (**a**) The difference between the simulated results and the predicted results; (**b**) the simulated results; and (**c**) predicted results by the MLNN-GABP neural network.

*5.4. Order of Importance of the Input Variables*

Sensitivity analysis was used to determine the thermal comfort evaluation for the MRT input factors. As illustrated in Figure 9, the positive axis indicates that the input variable is positively correlated with the output variable, whereas the negative axis indicates that the input variable is negatively correlated with the MRT distribution, and the larger the value, the greater the influence of that input variable on the MRT distribution. Thus, the order of importance for the input variables can be determined accordingly. The analysis of this importance rating can provide direction for future architectural design and environmental shaping. To begin, it was discovered that surface temperature and global radiation are the primary contributing elements, whereas the sky coefficient has a significant effect on building shape, and the building shadow is adversely connected with the MRT. Therefore, the proposed optimization measures for these four parameters will be more effective in future architectural and urban design.

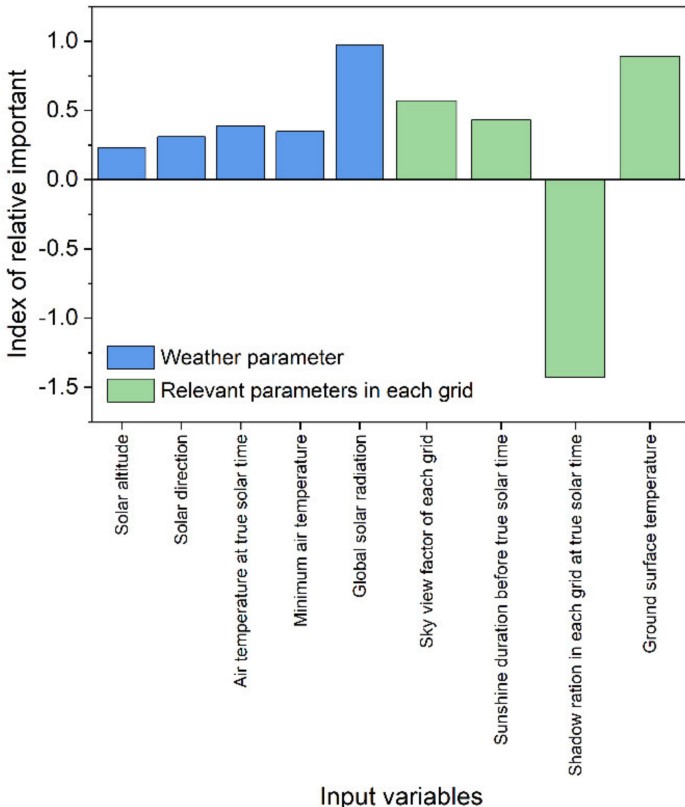

**Figure 9.** Order of importance of the input variables for hot summer days.

## 6. Conclusions

Using meteorological data and artificial neural networks (ANNs), we developed a unique technique of MRT prediction for a reduced block construction model. In addition, we compared the results of the ANN predictions with the results of the heat balance analysis simulations conducted during the years 2014–2018, as well as for the year 2019, which were not included in the training datasets but were acquired from other sources. The following are the findings and ramifications of this study.

1. We selected the weather parameters (solar altitude, solar direction, air temperature at noon true solar time, minimum air temperature, and global solar radiation) and the related parameters in each mesh (the sky view factor of each grid, sunshine duration before noon true solar time, the shadow ration in each grid at noon true solar time, and ground surface temperature) as the input parameters for the training datasets, which proved suitable for the MRT distribution prediction under a particular building form.

2. The article optimized the MLNN using a BP algorithm and GA; the optimization results indicated that the GA and BP algorithm have some effect on the results. The neural network without optimization produces more deviant prediction values, whereas the optimized neural network produces more stable prediction values when dealing with a large matrix.

3. An MLNN-GABP model was created to derive the MRT values from the weather variables during a specified period. After five years of training, it is possible to forecast the MRT distribution for the following year. The prediction results are extremely accurate, and a comparison of the MRT distribution plots reveals that they are consistent. As a result, the influence of this input parameter on the output can be analyzed using sensitivity analysis, and a relationship between the grid-related indicators and the MRT distribution can be determined.

Still, the proposed method has limitations. First, the MRT was only predicted using meteorology data for Sendai (38°16′06″ N, 140°52′10″ E), considering simplified building shapes, and the method only proposes an algorithm; further verifications are expected for realistic urban blocks in future works. It has not been validated in this study whether this method is applicable if other parameters are considered, such as trees, water, etc. Meanwhile, it is also important to consider wind patterns and non-realistic building arrangements. Therefore, further analysis will need to be conducted in subsequent research to address this issue. Second, the developed prediction method was only confirmed to predict the simulated results; it should be validated if this method can predict the MRT in real buildings. Finally, although this prediction has very high accuracy, it requires a large number of training datasets. Therefore, it would be very difficult to obtain five-year training data if this method were scheduled to be used in a real building environment and measured in the future.

## 7. Future Work

First, we will validate whether the method is accurate after considering trees, water bodies, grasses, etc. For this purpose, we will select a real building and build a similar model (including vegetation, water bodies, etc.), and adjust the input parameters of the grid accordingly. We will compare the accuracy of the method in predicting the MRT values and the order of importance for the input parameters in each grid for different building heights and different distances between the buildings to extend the applicability of the method. Second, if this method can accurately predict the distribution of the MRT around the model as the actual building, then we will combine this method with real measurements, where the surface temperature can be measured by drones, and weather data and other grid-related data can be calculated by measurements or software simulations. This is because MRT is more difficult to measure in real measurements, though the wind velocity, humidity, and air temperature can be more easily measured. The MRT distribution around the building can be quickly predicted by this method, and even the further prediction of the SET* distribution can be tried later. Finally, for a large number of training datasets, we will try to combine PCA and K-means and other methods in the future to remove weather conditions that are too similar, which helps to reduce the computation time and keep accurate prediction results while retaining high prediction accuracy. After completing these steps, we will be able to combine neural networks to analyze outdoor thermal environments very quickly and accurately, which will provide a reference for designers to propose strategies to adapt to urban warming and to improve outdoor thermal environments more efficiently.

**Author Contributions:** Conceptualization, Y.X.; methodology, Y.X.; software, Y.X.; validation, Y.X. and W.H.; formal analysis, Y.X. and W.H.; investigation, S.Y.; resources, C.L.; data curation, Y.X.; writing—original draft preparation, Y.X. and W.H.; writing—review and editing, Y.X. and W.H.; visualization, Y.X.; supervision, X.Z.; funding acquisition, Y.X. and X.Z. All authors have read and agreed to the published version of the manuscript.

**Funding:** This study was funded by the Fundamental Research Funds for the Central Universities (WUT: 2021IVA034), Hubei Provincial Natural Science Foundation (Grant No. 2021CFB005), the China Scholarship Council (No. 201708430100), and Humanities and Social Science Research Project of Wuhan Institute of Technology (WIT: 21QD36).

**Conflicts of Interest:** The funders had no role in the design of the study; in the collection, analyses, or interpretation of data; in the writing of the manuscript, or in the decision to publish the results.

## Abbreviations

List of symbols

| | |
|---|---|
| MRT | Mean Radiant Temperature |
| MLNN | Multi-Layer Neural Network |
| SVF | Sky View Factor |
| MLNN-GABP | MLNN optimized by the GA and BP algorithm |
| BPNN | Back-Propagation Neural Network |
| RNN | Recurrent Neural Network |
| $F$ | Fitness value |
| $Xi$min | minimum value in $Xi$ |
| $Yi$min | minimum value in $Yi$ |
| MAPE | Mean Absolute Percentage Error |
| GA | Genetic Algorithms |
| BP | Back-Propagation |
| ANN | Artificial Neural Network |
| SET* | Standard Effective Temperature |
| FFNN | Feed-Forward Neural Network |
| GA-BPNN | Back-Propagation Neural Network improved by the Genetic Algorithm |
| $Xi$max | maximum value in $Xi$ |
| $Yi$max | maximum value in $Yi$ |
| RMSE | Root Mean Square Error |

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
