# Peer review of "Artificial Neural Network Modeling for Predicting and Evaluating the Mean Radiant Temperature around Buildings on Hot Summer Days"

_buildings, doi:10.3390/buildings12050513_

Round 1

Reviewer 1 Report

The presented work must be seen as an introduction of an algorithm rather than a design method in urban planning. The application of the algorithm is worthy to investigate. But the case study is not realistic because of missing parameter like wind pattern and non realistic building arrangement.

I would recommend that the authors will state this fact in the text.

Here are some comments:

The simulation environment needs to be explained. Which software is used? Which type of software?

Which type of ANN is used?

Line 86: “including the NN ….” Please provide the full name.

Line 180: Is a time period of 4 years for the training data practical?

Figure 8: the scale needs labeling

Author Response

Reviewer #1:

We are grateful to Reviewer #1 for their valuable comments and useful suggestions that have helped us to improve our paper remarkably. As indicated in the responses that follow, we have taken these comments and suggestions into account in the revised paper. Please check the attached file and highlighted parts. Please feel free to contact us if there is any problem. Thank you very much!

Reviewer 2 Report

The presented article is interesting and suitable for Buildings magazine. It deals with a multilayer neural network optimized using genetic algorithms to predict and evaluate the mean radiant temperature around buildings on hot summer days. The article is written on 20 pages, applies knowledge from 78 literary sources, contains 5 tables, 9 pictures.

The results are important for the analysis of important input parameters influencing the mean radiation temperature on hot summer days - for days with the highest air temperature above 30 ° C. The basic idea was:

To predict the distribution of MRT around buildings based on the commonly used multilayer neural network (MLNN).

Weather data from 2014–2018 and related network indices were considered as input parameters for neural network training and the distribution of MRT around buildings was predicted in 2019.

This has important implications for optimization strategies for future building designers and urban designers to improve thermal condition around buildings.

The article has individual parts: 1. Introduction; 2. Overview of this study; Weather data collection; Evaluation model determination; 3. Methodology; Heat transfer analysis and MRT calculation; Multi-layer neural network ....; Sensitivity analysis approach; 4. The structure of the neural network Selecting the input parameters; The structure of MLNN-GABP; 5. Validation results; Validating the input parameters and number of hidden layer; Comparison of MLNN-GABP and MLNN; The prediction results of MRT distribution; Order of importance of the input variables; 6. Conclusions; 7. Future work;

I have comments on the post:

The title should be modified to make it more concise.

At the outset, it is clearly stated that, according to research findings, the increase in temperature is estimated to reach an increase of 1.5 ℃ between 2030 and 2050.

 See lines 95-152, it's full of all sorts of abbreviations and symbols.

There are basically a lot of abbreviations and explanations for the equations throughout the article. Therefore, insert a list of symbols and abbreviations at the beginning or end of the article. It will be easier for you to read. Otherwise it is confusing.

In table 1 corrections KJ / (m3 · K) to kJ / m3. K

I don't understand the composition of the roof. Is there plaster on the outer surface? - What? Describe in more detail!

Why is a 30 cm layer of soil considered for the floor? Is it sufficient for the calculation?

As early as 1987, the human model was identified as rectangular (according to Figure 3?). Have any other studies been conducted in this area since then?

Line 271 ... cl is the weighting factor? ... and l is 1,2,3 or -1, -2, -3;

Equations on lines 318 to 322 (indent more lines), this is confusing. Similarly, other lines 355, 356 or 368, 369 or 374, 375.

Add author contributions at the end of the article. Give specific abbreviations of the authors - not XX, XY, XZ ....

Conclusions can be accepted. A new MRT prediction method was developed for a simplified block building model using meteorological data and LV. Weather parameters and related parameters in each network were selected as input data for the training data sets to predict the MTR distribution in a particular building shape. In the article, certain values ​​were optimized and it was shown that the proposed algorithm has an effect on the results. A model was developed to derive MRT values ​​from weather variables over a specified time period. Although this prediction has a very high accuracy, it requires a large number of training data files and their verification by practical in situ measurements.

After the necessary adjustments, I support the publication of the article in the considered magazine.

Author Response

We are grateful to Reviewer #2 for their valuable comments and useful suggestions that have helped us to improve our paper remarkably. As indicated in the responses that follow, we have taken these comments and suggestions into account in the revised paper. Please check the attached file and highlighted parts. Please feel free to contact us if there is any problem. Thank you very much!

Reviewer 3 Report

In this work, the Authors proposed a new approach, based on a multilayer neural network (MLNN) optimized by genetic algorithms (GA) and backpropagation (BP) algorithms, to predict the distribution of mean radiant temperature around buildings. The methodology was applied considering a simplified urban layout in the climate of Sendai, Japan, for the period 2014-2019.

I found this contribution very well written, with a good overview of the state-of-the-art and a clear presentation of the methods. Although with some limitations, already clearly identified by the Authors who will deal with them in future developments, predicted values of mean radiant temperature were observed to be sufficiently accurate.

I have no particular remarks for the Authors and I recommend the acceptance of their manuscript for publication.

Author Response

Reviewer #3:

We are grateful to Reviewer #3 for their valuable comments and useful suggestions that have helped us to improve our paper remarkably. Thank you very much for your approval and encouragement of our research!

Round 2

Reviewer 1 Report

Accept in present form

Reviewer 2 Report

It can be seen that the authors approached the editing and correction of the manuscript responsibly. They corrected any deficiencies alleged by the reviewer. I just remind you: In Table 1, row 269 corrections KJ / m3.K ... kJ / m3.K The article is valuable, brings new results as well as outlines the direction of further research. I recommend publishing.